# Herpes Simplex Virus Type 1–Encoded miR-H2-3p Manipulates Cytosolic DNA–Stimulated Antiviral Innate Immune Response by Targeting DDX41

**DOI:** 10.3390/v11080756

**Published:** 2019-08-15

**Authors:** Yongzhong Duan, Jieyuan Zeng, Shengtao Fan, Yun Liao, Min Feng, Lichun Wang, Ying Zhang, Qihan Li

**Affiliations:** 1Department of Viral Immunology, Institute of Medical Biology, Chinese Academy of Medical Sciences & Peking Union Medical College, Kunming 650118, China; 2Experimental Center for Medical Research, Kunming Medical University, Kunming 650500, China

**Keywords:** antiviral immune response, cytosolic DNA sensor, HSV-1, miR-H2-3p

## Abstract

Herpes simplex virus type 1 (HSV-1), one of the human pathogens widely epidemic and transmitted among various groups of people in the world, often causes symptoms known as oral herpes or lifelong asymptomatic infection. HSV-1 employs many sophisticated strategies to escape host antiviral immune response based on its multiple coding proteins. However, the functions involved in the immune evasion of miRNAs encoded by HSV-1 during lytic (productive) infection remain poorly studied. Dual-luciferase reporter gene assay and bioinformatics revealed that Asp-Glu-Ala-Asp (DEAD)-box helicase 41 (DDX41), a cytosolic DNA sensor of the DNA-sensing pathway, was a putative direct target gene of HSV-1-encoded miR-H2-3p. The transfection of miR-H2-3p mimics inhibited the expression of DDX41 at the level of mRNA and protein, as well as the expression of interferon beta (IFN-β) and myxoma resistance protein I (MxI) induced by HSV-1 infection in THP-1 cells, and promoted the viral replication and its gene transcription. However, the transfection of miR-H2-3p inhibitor showed opposite effects. This finding indicated that HSV-1-encoded miR-H2-3p attenuated cytosolic DNA–stimulated antiviral immune response by manipulating host DNA sensor molecular DDX41 to enhance virus replication in cultured cells.

## 1. Introduction

Host cells have evolved a variety of mechanisms to counteract viral infection, especially interferon (IFN)-dependent antiviral innate immunity. The cytosolic DNA–stimulated innate antiviral immune signaling pathway is a new-type cell-intrinsic response to DNA viruses or stimuli of synthetic double–stranded DNA (dsDNA) analogs, which differs from Toll-like receptors (TLRs), RIG-I-like receptors, and NOD-like receptors signaling pathway. The sensor molecules of cytosolic DNA, such as DNA-dependent activator of IFN-regulatory factors (IRFs) (DAI) [1], gamma-IFN-inducible protein 16 (IFI16) [2], cyclic GMP-AMP (cGAMP) synthase (cGAS) [3], DEAD-box helicase 41 (DDX41) [4], and absent in melanoma 2 (AIM2) [5], are activated by exogenous DNA and recruit adaptor protein stimulator of IFN genes (STING) subsequently to trigger type I IFN production, leading to the expression of interferon–stimulated genes (ISGs) and inflammatory cytokines, resulting in antiviral effects [6].

Herpes simplex virus type 1 (HSV-1) employs many sophisticated strategies to escape host antiviral immune response depending on protein-based or microRNA (miRNA)-based mechanisms during long-time coevolution between hosts and viruses. HSV-1-encoded infected-cell polypeptide 0 (ICP0) is an immediate early (IE, α) protein possessing an E3 ubiquitin ligase activity, expressing highly during lytic infection. It promotes the degradation of certain host proteins including alpha thalassemia/mental retardation syndrome X-linked (ATRX) [7], myeloid differentiation factor (MyD88), MyD88 adaptor–like protein (Mal), and Toll-interleukin 1 receptor (TIR) domain–containing adaptor protein (TIRAP) [8], P65 [9], IFI16 [10,11,12], and STING [13] to enhance viral infection and replication by abrogating type I IFN production and facilitating immune evasion [14]. HSV-1 virion host shutoff (VHS) protein encoded by *UL41* gene takes on a substrate-specific endonuclease activity, with function similar to that of RNase [15], which attenuates the expression and activity of antiviral proteins including viperin, IFIT3, and tetherin to boost viral replication by cleavage of its mRNAs for degradation [16,17,18,19]. In addition, some other viral proteins such as UL36USP [20,21,22,23], US11 [24,25], ICP34.5 [26,27], US3 [28,29,30], ICP47 [31,32], ICP27 [33], and so on are proved to be involved in immune evasion by targeting correlative components of the antiviral immune pathway.

miRNAs are endogenous ~22-nucleotide (nt) small noncoding RNAs that can play important regulatory roles in animals and plants by targeting mRNAs for cleavage and translational repression at the posttranscriptional levels [34]. Many cellular miRNAs play a key role in the posttranscriptional regulation of almost every cellular gene regulatory pathway, including the fine-tuning of immunological circuits, and therefore it is not surprising that viruses have found ways to subvert this process using virally encoded miRNAs [35,36]. The first virally encoded miRNAs (miR-BHRF1 and miR-BART) were identified in Epstein–Barr virus (EBV) [37]. After that, more miRNAs encoded by the herpesvirus family have been discovered and characterized [38,39,40,41]. HSV-1 encodes 18 viral precursor miRNAs (pre-miRNAs) and 27 mature miRNAs (miRBase, release 22.1: October 2018), clustered mainly in the region of the latency-associated transcript (LAT) locus, which is the only abundant transcriptional noncoding region during latent infection [42,43,44,45,46]. Previous studies showed that HSV-1-encoded miR-H1-3p attenuated the repression of lytic viral gene expression by targeting ATRX directly, components of nuclear domain 10 (ND10) localizing adjacent to incoming viral genomes and generating a repressive environmental for viral gene expression [7]. HSV-1-encoded miR-H2-3p, miR-H4-5p, and miR-H6-3p target viral ICP0, ICP34.5, and ICP4 mRNA, respectively, to regulate viral latency and virulence [43,47]. HSV-1-encoded miR-H8 modulates the glycosylphosphatidylinositol (GPI) anchoring pathway, which is associated with ligands for nature killer (NK) cells activating receptors, through targeting GPI transamidase component PIG-T (PIGT) as GPI components and suppressing the expression of tetherin to evade elimination by NK cells and type I IFN pathway [48]. HSV-1-encoded miR-H27 inhibits a cellular transcriptional repressor of viral IE and early genes by blocking the expression of KLHL24, evading host cell defenses, and supporting the efficient replication [49]. HSV-1-encoded miR-H28 and miR-H29 were expressed late in productive infection and exported from infected cells via exosomes. The ectopic expression of miR-H28 and miR-H29 mimics in human cells before infection reduced the accumulation of viral mRNA and proteins, and reduced plaque sizes and viral yields. These results showed that the main purpose was to restrict HSV-1 replication and spread in recipient cells [50]. Despite some progress in aspects of functional research of HSV-1 miRNAs in recent years, most functions of the HSV-1 miRNAs are largely unknown, especially involving the regulation of immune responses to HSV-1 infection. 

This study was performed to screen, identify, and characterize HSV-1-encoded miR-H2-3p as a suppressor of the cytosolic DNA–stimulated antiviral innate immune pathway by targeting DNA sensor DDX41 to counteract the production of type I IFN and reinforce HSV-1 immune evasion. 

## 2. Materials and Methods 

### 2.1. Cells and Cell Culture 

Human embryonic kidney 293T (HEK-293T) cells and human leukemic monocyte (THP-1) cells were obtained from the Institute of Medical Biology (Kunming, China) and propagated in Dulbecco′s modified Eagle′s medium (DMEM, Thermo Fisher Scientific, MA, USA), PRMI 1640 (Gibco, MA, USA), respectively, supplemented with 10% (*v*/*v*) fetal bovine serum (FBS, Gibco) and penicillin–streptomycin (100 U/mL and 100 µg/mL). African green monkey kidney (Vero) cells obtained from the Institute of Medical Biology and human foreskin fibroblast (HFF) cells were purchased from Cell Resources Center, Shanghai Academy of Sciences, Chinese Academy of Sciences (Shanghai, China) and cultured in minimum Eagle′s medium (MEM, Thermo Fisher Scientific, MA, USA), supplied with 10% FBS and penicillin–streptomycin (100 U/mL and 100 µg/mL). The cells were maintained at 37 °C in a humidified atmosphere containing 5% CO_2_.

### 2.2. Virus and Viral Titer Assay 

HSV-1 (F strain) was obtained from the Institute of Medical Biology and propagated in Vero cells at a multiplicity of infection (MOI) of 0.01. The virus-infected cells were harvested at the peak of cytopathogenic effect (CPE), followed by three-time freezing and thawing. The supernatant was collected, centrifuged, titrated by cell culture infective dose 50% (CCID50) method, and then stored at −80 °C for further assay. 

### 2.3. Transfection and Infection Assay

HEK-293T cells or THP-1 cells were seeded in 24 well plates (5 × 10^5^ cells/well) 24 h before infection. Then, the cell medium was removed, and fresh antibiotic-free opti-MEM medium (Thermo Fisher Scientific, MA, USA) was added. Transfection with plasmid or HSV-1 60-mer dsDNA was performed with Lipofectamine 2000 reagent (Invitrogen, CA, USA) according to the manufacturer′s protocol. The cell culture medium was replaced with fresh medium after 6 h, and the cells were cultured for another 24 h before HSV-1 infection at indicated MOI. The cells were harvested at different time points, and further experiments were carried out. 

### 2.4. Plasmid Construction

Dual-luciferase reporter plasmid pmirGLO (Promega, WI, USA) carrying human *DDX41, IFI16, DAI, cGAS, STING, TRIF, TBK1, TLR9, IRF3, IRF7,* and *IRF9* mRNA 3′ untranslated region (3′-UTR) full-length sequences was constructed with *Sac*I and *Xba*I restriction enzyme sites using the standard molecular clone protocol, and the correctness was verified by sequencing according to reference sequences in NCBI. The full-length 3′-UTR sequences of these genes were amplified by PCR using double-enzyme cleavage site primers from HEK-293T cell genomic DNA, and served as templates in PCR. The primers used for plasmid construction are listed in Table 1 (underlining indicates the cleavage site of *Sac*I in the forward primer and *Xba*I in the reverse primer):

For constructing positive control plasmid, oligo DNA sequences were also commercially synthesized, including human miR-16 and HSV-1 miR-H2-3p complete complementary sequences, and cloned into pmirGLO using *Sac*I and *Xba*I restriction enzyme sites. HSV-1-encoded DNA polymerase *UL30* was employed to construct a standard plasmid named pMD18-T-UL30 using a pMD^TM^18-T Vector Cloning Kit (TaKaRa Bio, Dalian, China). The *UL30* sequence of 92 bp was amplified from HSV-1 genome DNA. 

### 2.5. Site-Directed Mutagenesis

Site-directed mutagenesis was performed using a mutagenesis kit according to the manufacturer’s protocol (Tiangen, Beijing, China). The mutant primers were as follows (underlining indicates the mutant sites):

*DDX41* mutant forward: 5′-ACAGAATCAGCATTTCACCACACCTGGCCTGGAATGGGCC-3′

*DDX41* mutant reverse: 5′-GGCCCATTCCAGGCCAGGTGTGGTGAAATGCTGATTCTGT-3′

### 2.6. Dual-Luciferase Reporter Assay

HEK-293T (5 × 10^5^ cells/well) in 24 well plates were transfected with a reporter plasmid (500 ng) or control plasmid (500 ng) without inserting exogenous sequences of empty pmirGLO. After 24 h, the cells were infected with HSV-1 or underwent mock treatment for 24 h. Firefly and Renilla luciferase activities were measured consecutively with the Dual-Luciferase Assay System (Promega, WI, USA) 48 h following transfection using a BioTek synergy 4 multifunctional reader (BioTek, VT, USA). Firefly luciferase activity was normalized to Renilla luciferase activity, and then normalized to the average activity of the control reporter.

### 2.7. RNA Interference

Specific knockdown of DDX41 or IFI16 in THP-1 cells was induced with artificial synthetic small interfering RNA (siRNA) duplexes (GenePharma, Shanghai, China). Transfection was performed with Lipofectamine 2000 reagent (Invitrogen, CA, USA). The silencing efficiency was evaluated by immunoblot or qPCR analysis in THP-1 cells. The siRNAs sequences are listed in Table 2.

### 2.8. miRNA Mimics and Inhibitors 

miR-H2-3p mimics, mimics’ negative control (double-stranded RNA oligonucleotides) and inhibitors, and inhibitors’ negative control (single-stranded RNA oligonucleotides) (GenePharma, Shanghai, China) were transfected into HEK-293T cells using Lipofectamine 2000 at different final concentrations. The miRNA mimics and inhibitor sequences are listed in Table 3.

### 2.9. RNA Quantification

Total RNA was extracted from HEK-293T, HFF, or THP-1 cells with TRIzol-A^+^ reagent (Tiangen, Beijing, China) following the manufacturer′s protocols. Quantitative reverse–transcription polymerase chain reaction (RT-qPCR) assay was performed using Bio-Rad CFX96 (Bio-Rad, CA, USA) and PrimeScript RT reagent kit with a gDNA eraser (TaKaRa Bio, Dalian, China). The RT reaction system comprised 0.25 μM gene-specific RT primer, RT enzyme mixture, and total RNA erased genomic DNA in a total volume of 20 µL. Each 25 μL reaction system of qPCR contained 0.5 μM of each PCR primer, 12.5 μL of SYBR Green mix, 1 μL of RT reaction solution, and RNase-free water. The data were normalized to the expression of glyceraldehyde-3-phosphate dehydrogenase (GAPDH) in each individual sample, and the 2^−ΔΔCt^ method was used to calculate changes in relative expression. Gene-specific primers used for RT-PCR assays are listed in Table 4. 

Quantification of mature miRNAs was carried out based on the stem-loop RT-qPCR assays [51]. The detailed protocol was as described earlier. The relative expression of HSV-1 mature miR-H2-3p was normalized to that of internal control U6 small nucleolar RNA (snoRNA) within each sample using the 2^−ΔΔCt^ method, and then standardized to the miRNA level in mock-infected or control miRNA-treated cells. 

### 2.10. QPCR Detection of Viral Load 

The cell culture supernatant was collected, and viral genome was extracted for determining viral load using an AxyPrep body fluid virus DNA/RNA extraction kit (Axygen, Jiangsu, China) according to the specificity. The absolute quantification of gene *UL30* copies was performed and normalized to standard plasmid pMD18-T-UL30.

### 2.11. Immunoblotting

THP-1 cells were lysed in Western and immunoprecipitation (IP)cell lysis buffer (Beyotime, Shanghai, China), and total protein concentrations were determined using a Bradford protein assay kit (Beyotime). Then, 40 µg of total protein per lane was separated on 10% Tris-glycine sodium dodecyl sulfate–polyacrylamide gel electrophoresis (SDS-PAGE) gels and transferred onto 0.45 μm polyvinylidene difluoride (PVDF) membranes. This was followed by blocking in 5% nonfat milk powder in Tris-buffered saline/Tween. The blots were probed with primary antibodies to human DDX41 (polyclonal antibody, Absin, China), IFI16 (polyclonal antibody, Absin), and GAPDH (polyclonal antibody, Absin), and then probed with horseradish peroxidase-conjugated goat anti-rabbit IgG (H + L) (polyclonal antibody, Absin). The bound secondary antibodies on PVDF membranes were detected by a BeyoECL Plus kit (Beyotime) using a Bio-Rad ChemiDoc XRS+ instrument (Bio-Rad, CA, USA). The relative density of immunoblotting bands was determined using ImageJ software (NIH, USA) according to the manual.

### 2.12. Bioinformatics Analysis

The candidate genes that could be targeted by HSV-1-encoded miRNAs were predicted using RNAhybrid algorithms [52,53]. Predictions were ranked based on the minimum free energy (MFE), evaluated using the RNAhybrid program.

### 2.13. Statistics and Data Analysis

All results were presented as means ± standard deviations (SD). The groups were compared using the two-tailed unpaired Student *t* test, as marked by asterisks in some of the figures. Data were statistically analyzed using GraphPad Prism 7.0 (GraphPad Software, CA, USA).

## 3. Results

### 3.1. HSV-1 Infection Attenuated the Cytosolic DNA Sensor Reporter Expression of DDX41

The involvement of miRNAs from host cells and/or HSV-1-encoded miRNAs during viral infections in the regulation of the cytosolic DNA–stimulated type I IFN pathway was investigated by constructing a dual-luciferase reporter carrying a 3′-UTR of signaling protein mRNAs, which included cytosolic DNA sensor molecules IFI16, DDX41, cGAS, and DAI; adaptor molecules STING and TRIF; and other key member molecules IRF3, IRF7, IRF9, TLR9, and TBK1 (Figure 1A).

Although miRNAs have multiple posttranscriptional regulatory mechanisms, the most typical mechanism of action is to regulate gene expression, including translational repression and degradation via interaction with the 3′-UTR of mRNA [34], which is the basis for many software for miRNA target gene prediction. Regardless of lytic infection in vitro or latent infection in neurons in vivo, a large number of virally encoded miRNAs were expressed by HSV-1, especially productive infection at a late time point [42,46]. Hence, the generation of typical CPE occurrence with an MOI of 1 at 24 h postinfection (hpi) was chosen for the expression of viral miRNAs, and the function of blocking the reporter expression was screened (Figure 1B). The next screening of the virus-encoded miRNAs for the potential involvement in the cytosolic DNA-stimulated type I IFN pathway showed that the reporter expression of bearing a DDX41 mRNA 3′-UTR sequence decreased significantly in HEK-293T and THP-1 cells. Some others, such as TRIF and DAI reporter expression had a significant decrement only in THP-1 cells (Figure 1C,D). Taken together, these results showed that DDX41 reporter expression was inhibited significantly in a 3′-UTR-dependent manner during HSV-1 infection.

### 3.2. HSV-1-Encoded miR-H2-3p Targeted DDX41 Directly

RNAhybrid software was employed to further investigate the miRNAs that could target DDX41, especially focusing on the HSV-1-encoded one. Previous studies proved that the ability of miRNA to translationally repress a target mRNA was largely dictated by the free energy of binding sites from 2 to 7/8 nucleotides in the 5′ region of miRNAs. Meanwhile, G:U wobble base-pairing in this region interfered with activity [54]. Based on these rules, it was found that virally encoded miR-H2-3p was the best one fitted with the aforementioned region, although the MFE was not the smallest one in the prediction targets of miRNAs (Figure 2B and Appendix A).

The seed sequence of miR-H2-3p (from second to seventh or eighth nucleotides in the 5′ region of miRNAs) complementary binding sites to the DDX41 3′-UTR region is relatively conserved among humans, chimps, cows, and rhesus (Figure 2A). It is usually considered that conservation across species is one of the best parameters for predicting the targets of miRNAs. A three-base-pair mutation was introduced into the miR-H2-3p binding sites within the 3′-UTR of DDX41 mRNA (Figure 2B and Appendix A). Next, the luciferase reporter plasmid bearing the wild-type 3′-UTR of DDX41 mRNA or a three-base-pair mutation was co-transfected into HEK-293T with miR-H2-3p mimics or NC, respectively. It was found that miR-H2-3p mimics significantly decreased the expression of luciferase reporter plasmid bearing the wild-type 3′-UTR of DDX41 mRNA. However, the expression of luciferase reporter plasmid bearing a three-base-pair mutation was restored partially (Figure 2C). To examine the function of miR-H2-3p in vitro by gain-of-function methods, HEK-293T cells were transfected with miR-H2-3p mimics or NC at indicated final concentrations. The miR-H2-3p mimics increased the miR-H2-3p expression in a dose-dependent manner, as determined by stem-loop qRT-PCR (Figure 2D). To confirm the specificity of qRT-PCR-amplified products, agrose gel electrophoresis was performed, and the results showed that the band was consistent with the expected size (Figure 2E) Further experiments showed that the transfection of miR-H2-3p decreased the DDX41 expression at mRNA and protein levels in HEK-293T cells in a dose-dependent manner, suggesting that DDX41 expression could be regulated by miR-H2-3p via both mRNA cleavage and translational suppression (Figure 2F–H). Taken together, these experimental results indicated that miR-H2-3p could regulate DDX41 expression directly.

### 3.3. miR-H2-3p Contributed to the Downregulation of DDX41 during HSV-1 Infection in THP-1 Cells

To further verify the regulatory effect of endogenous miR-H2-3p generated during HSV-1 infection on DDX41 protein, THP-1 cells were infected with HSV-1 in different time periods or with different MOI, and then the miR-H2-3p expression was determined using qRT-PCR. At an MOI of 1, the expression of miR-H2-3p increased in a time-dependent manner (Figure 3A), indicating that this was an miRNA expressed in late infection. As for the MOI, the miR-H2-3p expression increased in a MOI-dependent manner, as indicated at 24 hpi (Figure 3B). For the assay of loss-of-function, THP-1 cells were transfected with miR-H2-3p-complementary synthetic oligonucleotides inhibitor or a negative control inhibitor at indicated final concentration, and then infected with HSV-1. The miR-H2-3p expression decreased by about 40% with 50 nM or by 70% with 100 nM inhibitor compared with negative control treatment (Figure 3C). The aforementioned experimental results showed that miR-H2-3p was abundantly expressed in THP-1 cells during HSV-1 infection, and miR-H2-3p inhibitor suppressed the expression effectively, which was a basis for the next experiments.

Previously accumulated research data indicated that DDX41 was induced during ubiquitination and degradation via K-48-mediated linkage by TRIM21, which is encoded by ISG and linked to the negative feedback regulation of type I IFN responses induced by cytosolic DNA [55]. Based on this evidence, the degradation of DDX41 protein in THP-1 cells was identified during HSV-1 infection. A significantly negative linear correlation was found between the expression of the miR-H2-3p and the expression of DDX41 protein (Figure 3D). Meanwhile the expression of DDX41 protein decreased gradually with the progress of HSV-1 infection at an MOI of 1 (Figure 3E,F). These results were consistent with the reported findings. Subsequently, the cells were transfected with miR-H2-3p mimics or NC and then infected with HSV-1 at an MOI of 1. With the progress of transfection and infection time, the DDX41 protein was more greatly reduced compared with individual infection (Figure 3G,H,K). However, treatment with miR-H2-3p inhibitor obviously reversed this trend compared with transfection with miR-H2-3p mimics and infection (Figure 3I–K). The expression of DDX41 protein was not significantly different between miR-H2-3p mimics and miR-H2-3p inhibitor at 72 hpi. It is believed that this might be the result of very low expression of DDX41 protein (Figure 3K). In summary, endogenous miR-H2-3p or exogenous miR-H2-3p mimics and inhibitor could regulate the expression of DDX41 directly in THP-1 cells.

### 3.4. DDX41 was a Crucial DNA Sensor in THP-1 Cells

IFI16 and DDX41 have been reported as DNA sensors in different types of cells, mediating the production of type I IFN via the DNA-stimulated pathway in response to cytosolic dsDNA [2,4]. Which DNA sensor molecular played a more important role for the induction of IFN-β by stimulation in THP-1 cells was not known. HFF cells that constitutively expressed IFI16 protein and were often used in HSV-1 infection served as a control for comparing the expression of IFI16 and DDX41 [56]. Significantly high expression of DDX41 was observed in THP-1 cells compared with that in HFF cells at the mRNA and protein levels. However, the expression of IFI16 showed an opposite trend in HFF cells (Figure 4A–C). To investigate the ability to respond to exogenous dsDNA mediated by IFI16 and DDX41, THP-1 cells were transfected with HSV-1 60-mer dsDNA along with specific siRNA for IFI16 or DDX41 knockdown or siRNA control. The levels of IFI16 and DDX41 protein significantly decreased 24 h after transfection in siRNA-treated cells, as detected by immunoblotting (Figure 4C). The induction of IFN-β by HSV-1 60-mer dsDNA significantly decreased after being treated with siRNA for DDX41 compared with that of IFI16 (Figure 4D). Together, these results indicated that DDX41 was an important DNA sensor in THP-1 cells. It might play a major role in the early stage of viral infection and serve as an ideal target for the action of viral miRNAs to escape the host immune system.

### 3.5. miR-H2-3p Negatively Regulated HSV-1-Triggered IFN-β and ISG Expression by Targeting DDX41

It is well known that HSV-1 infection triggers the expression of IFN-β and its downstream ISGs, which play a central role in the host antiviral innate immune response. To verify the roles of miR-H2-3p and DDX41 in the production of IFN-β and induced expression of ISGs during HSV-1 infection in THP-1 cells, siRNA was used to inhibit the expression of DDX41 protein, which was verified by immunoblotting (Figure 5A). The transfection of miR-H2-3p mimics or specific knockdown of DDX41 significantly decreased the expression level of IFN-β and MxI mRNA in the presence of HSV-1 (Figure 5B,C). Meanwhile, the transfection of miR-H2-3p was inhibited by the specific knockdown of DDX41 (Figure 5D,E). That is to say, the increased IFN-β and MxI expression due to miR-H2-3p inhibition was suppressed by the specific knockdown of DDX41. Similarly, the decreased IFN-β and MxI expression due to miR-H2-3p overexpression produced similar effects by the specific knockdown of DDX41. These data suggested that the knockdown of DDX41 phenocopied the attenuated IFN-β pathway effect of miR-H2-3p overexpression and counteracted the effect of miR-H2-3p inhibition. Together, these results showed that HSV-1-encoded miR-H2-3p negatively regulated the type I IFN pathway via the suppression of endogenous DDX41 expression.

### 3.6. miR-H2-3p Promoted HSV-1 Replication and Viral mRNA Expression in THP-1 Cells

To evaluate the effects of miR-H2-3p on HSV-1 replication, THP-1 cells were transfected with miR-H2-3p mimics or NC and then infected with HSV-1 at an MOI of 1. The samples were harvested at 6, 12, 24, 48, and 72 hpi. The levels of viral titer in the culture supernatant were determined by the CCID_50_ method, and the intracellular HSV-1 genome copy numbers were determined by absolute qPCR. The significant increases in the infectious virus yield were observed after miR-H2-3p mimics treatment compared with NC at 12, 24, 48, and 72 hpi, besides 6 hpi (Figure 6A). Accordingly, the intracellular viral genome absolute copy numbers significantly increased after miR-H2-3p mimics treatment compared with NC treatment at 6, 12, 24, 48, and 72 hpi, based on the standard curve with an amplification efficiency of 115% and a square of reliability coefficient of 0.9976 (Figure 6B,C). To evaluate the effects of miR-H2-3p on HSV-1 thymidine kinase (TK), glycoprotein C (gC), and ICP0 expression, THP-1 cells were co-transfected with miR-H2-3p mimics and NC at various concentrations, as indicated, and then infected with HSV-1 at an MOI of 1 for 24 h. The viral gene expression at the mRNA level was determined by qRT-PCR. TK and gC expression increased in a dose-dependent manner, whereas ICP0, which was proved as a target of miR-H2-3p and miR-138, showed an opposite trend compared with TK and gC (Figure 6D), which was consistent with previous findings [43,57].Together, these results showed that miR-H2-3p promoted HSV-1 replication and viral mRNA expression in THP-1 cells.

## 4. Discussion

Viruses have evolved multiple mechanisms to fight against the host antiviral defense system to ensure the integrity of their entire life cycle, just as the host evolved a variety of antiviral mechanisms, especially for HSV-1. Recent studies demonstrated that HSV-1-encoded miR-H2-3p, which is abundantly expressed in infected THP-1 cells, could inhibit the expression of DDX41 mRNA and protein, a pivotal DNA sensor molecular involved in the regulation of the typeIIFN pathway by DNA virus infection, and lead to the reduced expression of IFN-β and its downstream stimulated gene *MxI*, ultimately promoting viral replication. Many virally encoded proteins have been found to regulate the function of these DNA sensors, such as IFI16 by ICP0 and VHS [11,58], STING by ICP27, γ_1_ 34.5 and UL46 [33,59,60], cGAS by UL37, UL41, and VP22 [61,62,63], and AIM2 by VP22 [64]. However, this was the first screening and identification of HSV-1-encoded miRNA manipulating the cytosolic DNA–stimulated antiviral innate immunity pathway by targeting DDX41 mRNA 3′-UTR directly.

Transfection with dual-luciferase reporter plasmid-linked full-length 3′-UTR of selected genes, and then infection with HSV-1 F strain were performed to screen the decreased activity of the reporter, which might be caused by virally encoded miRNAs. Some special attention should be paid to the reduction of reporter activity caused by nonvirally encoded miRNAs or non-miRNAs. Numerous well-established studies revealed that HSV-1 infection gave rise to a change in miRNA expression profiles in host cells [65,66], producing nonvirally encoded miRNAs, targeting the reporter and causing reduced activity. More importantly, the HSV-1 *UL41* gene-encoded VHS protein had a similar mode of action as that of miRNA. It could degrade AU-rich mRNA in 3′-UTR mediated by RNA-binding protein tristetraprolin (TTP) [67,68]. For this reason, the candidate genes initially screened must be carefully and rigorously verified as targets of miRNAs, meanwhile excluding nonspecific effects. Intriguingly, innate immune-related genes were found to have much shorter 3′-UTR compared with the human protein-encoded genes (Figure 1A, in the box on the right); the average length of the 3′-UTR was approximately 800 nucleotides [69]. The shorter 3′-UTR implied fewer regulatory sites of miRNAs, although the regulatory sites of miRNAs were related not only to the full-length sequences, but also to the sequences of bases. Consistent with this assumption, a study using predictive algorithms showed that upstream factors regulating immunity, such as ligands and receptors (cytokines, chemokines, and TLRs), were nontargets in general [70]. Sequence analysis found that a large set of genes involved in basic cellular processes had short 3′-UTRs, specifically depleted of miRNA-binding sites [71]. This might be a basic strategy for maintaining internal system stability. Nevertheless, a recent study showed that MAVS—mitochondrial antiviral signaling protein, also known as IPS-1 (IFN-β promoter stimulator 1), CARDIF (CARD-adaptor-inducing IFN-β), or VISA (virus-induced signaling adaptor)—located in the mitochondrial outer membrane adaptor protein and involved in the antiviral innate immune response, had a longer than 9 kb of 3′-UTR. Three AU-rich elements (AREs), four hsa-miR-27a-binding sites, and a subcellular localization signal are contained in 3′-UTR, MAVS expression regulation occurs precisely at the posttranscriptional level via the long 3′-UTR [72].

HSV-1-encoded miR-H2-3p is located in the LAT region and is antisense to the ICP0 transcript. It is expressed in both latent infection in vivo and lysis infection in vitro [43]. Previous studies of miR-H2-3p function were performed mainly in HSV-1 mutant strains, yielding inconsistent results. For example, HSV-1 17syn+- and Mckrae strain-inactivated endogenous miR-H2-3p were capable of upregulating ICP0 mRNA and protein expression in infected 293T cells and ICP0 protein expression in rabbit skin cells. 17syn+ mutant strain replication was repressed in Neuron 2A cells, while the Mckrae mutant strain showed no significant difference in replication ability in mouse eye infection but showed an increment of toxicity and reactivation in animal experiments [47,73,74]. The KOS mutant strain did not increase ICP0 protein expression in infected Vero and Neuron 2A cells and showed the same ability of replication as its parental strain [75]. The ectopic co-expression of miR-H2-3p and ICP0 in 293T cells gave rise to the specific inhibition of ICP0 protein, but the ability to inhibit ICP0 mRNA expression was inconsistent in these studies [43,75]. Summarizing the existing research results proved that miR-H2-3p could inhibit ICP0 expression at both protein and mRNA levels via miR-H2-3p mutant inactivation and overexpression experiments in many types of cells in general. The experimental results also demonstrated that the transfection of miR-H2-3p could decrease the expression of ICP0 mRNA in THP-1 cells (Figure 6D, upper). The effect of miR-H2-3p on viral replication exhibited specificity of cell type and viral strain type, which might explain the complexity of its mechanism of action to a certain degree. The present study proved that miR-H2-3p could repress DDX41 mRNA and protein expression directly and promote HSV-1 F strain replication and gene (*TK* and *gC*) expression in late-infected THP-1 cells (Figure 6). This might throw light on a novel evasion of innate antiviral response based on miRNA by HSV-1. It is worth mentioning that neuron-specific miR-138 can repress ICP0 expression more effectively compared with miR-H2-3p [57]. This functional redundancy of miR-H2-3p may be a footnote for its other regulatory functions. The interaction between miR-H2-3p and ICP0 and their mechanism of regulating HSV-1 replication still need to be further studied.

DDX41 is a member of the DEXDc family of RNA helicases. It is characterized by the conserved motif Asp-Glu-Ala-Asp (DEAD) and identified as an intracellular DNA sensor in myeloid dendritic cells [4]. DDX41, phosphorylated and activated by Bruton’s tyrosine kinase, can recognize and bind intracellular dsDNA, cyclic-di-GMP (c-di-GMP), or cyclic di-CMP (c-di-CMP). It complexes with STING to signal to TBK1-IRF3 and activate type I IFN signaling [76,77]. As mentioned earlier, TRIM21 negatively regulates the antiviral innate immunity responses by inducing ubiquitination and degradation of DDX41 at the post-translational level. The present study provided evidence that DDX41 was a target gene for miR-H2-3p and was downregulated at the posttranscriptional level by HSV-1 infection in THP-1 cells. HSV-1 adopted virally encoded miR-H1 and ICP0 to repress the expression of ATRX, an effector of intrinsic immunity [7]. Why HSV-1 used different mechanisms to regulate the expression of the same protein was worth thinking. Perhaps this multilevel regulatory mechanism could more effectively promote immune evasion compared with a single RNA-base or protein-base evasion mechanism. It is also a manifestation of the complexity and diversity of HSV-1 evasion mechanism.

In addition to determining the accumulation of viral ICP0 mRNA, expressed as an IE gene and served as a transcriptional activator to promote viral replication [78,79], we also determined the accumulation of mRNAs of representative early (β) gene *TK* and late (γ) gene *gC*. miR-H2-3p promoted *TK* and *gC* expression and viral replication, while inhibiting ICP0 expression. This may reveal that the contribution of ICP0 to viral replication was very small in the late stage of viral infection (24 hpi), even if ICP0 inhibited by miR-H2-3p did not significantly affect viral replication. By way of comparison, miR-H2-3p promoted viral replication and the expression of early and late genes mainly by manipulating DNA-stimulated antiviral immune signaling pathway in the late stage of infection.

The present study revealed a new immune evasion mechanism through which HSV-1-encoded miR-H2-3p targeted DNA sensor DDX41 mRNA 3′-UTR directly to subvert the type I IFN pathway and escape host elimination (Figure 7). A significant difference from the virus-encoded proteins involved in immune evasion was that miRNA regulation occurred at the posttranscriptional level, causing degradation or cleavage or translational inhibition of the target mRNA, while target genes were regulated via ubiquitination or de-ubiquitination, phosphorylation or dephosphorylation, methylation or demethylation, and SUMOylation or de-SUMOylation. Most studies suggested regulation at the posttranscriptional level as a cost-effective way.

## Figures and Tables

**Figure 1 viruses-11-00756-f001:**
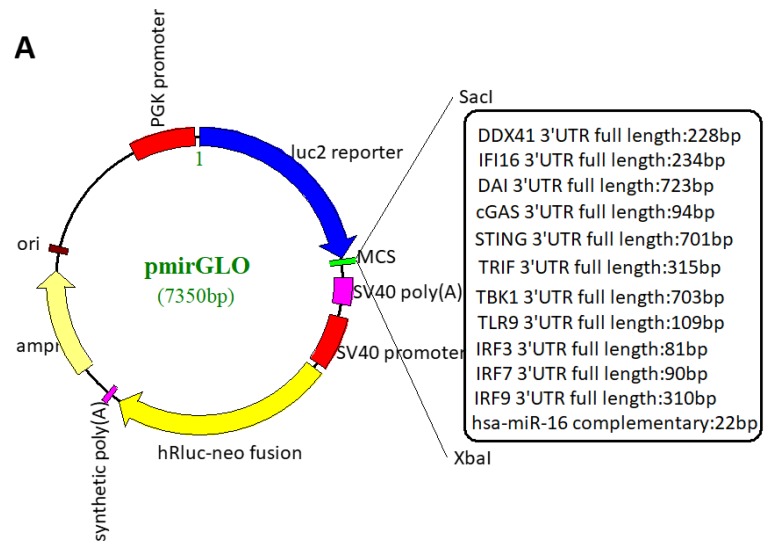
HSV-1 infection attenuated cytosolic DNA sensor reporter expression. (**A**) The scheme of dual-luciferase reporter plasmid construction. pmirGLO vector was digested with *Sac*I and *Xba*I, and then inserted into full-length 3’-UTR sequences of different mRNAs with a correct linker connected with T4 DNA ligase. Hsa-miR-16 complement (pmirGLO-16), as a positive internal control, was constructed by cloning complete complementary sequences into pmirGLO vector. (**B**) Different CPEs of HKE-293T cells infected with HSV-1 at different viral titers and times. HEK-293T cells were infected with HSV-1 at an MOI of 0.1, 1, 10, and 50 or Mock and snapped at the indicated time points. Scale bar, 100 µm. (**C**,**D**) HEK-293T (**C**) or THP-1 (**D**) cells were transfected with a reporter plasmid, and then infected with HSV-1 or mock. The cells were transfected with 500 ng indicated reporter plasmid. After 24 h of transfection, the cells were infected with HSV-1 at an MOI of 1 or mock for another 24 h, collected, lysed, and assayed for dual-luciferase activity. The values present the percentage of reporter activity with infection compared with that with mock infection, standardized to 100% in the empty vector (EV) group. Data are the means ± SD (*n* = 3) from one representative experiment. Similar results were obtained from three independent experiments. * *P* < 0.05; ** *P* < 0.01; *** *P* < 0.001.

**Figure 2 viruses-11-00756-f002:**
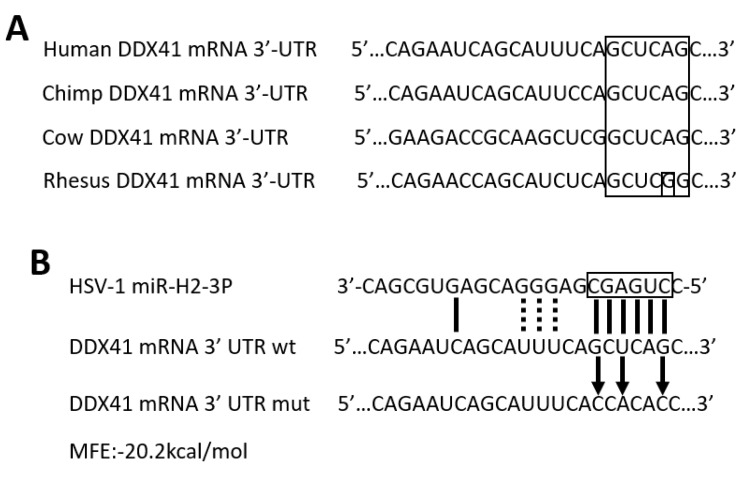
miR-H2-3p targeted DDX41 directly. (**A**) Sequence alignment of DDX41 from different species for conservative analysis. The seed region binding within DDX41 in the big black rectangular frame, especially G:U wobble base pairs in the small black rectangular frame. (**B**) Sequence alignment of miR-H2-3p and its binding sites in the 3′-UTR of DDX41. The seed sequence in the 5′ region of miR-H2-3p is shown in the black rectangular frame, pairing nucleotides in black short sticks or black short dashed line (only for G:U wobble base pairs), mutant sites in black short arrows. (**C**) HEK-293T cells were co-transfected with empty plasmid (EV), pmirGLO-DDX41-3′-UTR wild-type (DDX41 wt) plasmid, pmirGLO-DDX41-3′-UTR mutation (DDX41 mut) plasmid, and miR-H2-3p mimics (miR-H2-3p) or NC as indicated. After 24 h of transfection, the cells were harvested and dual-luciferase reporter assay was performed. The values were standardized to 100% in the EV + NC groups. Data are the means ± SD (*n* = 3) from one representative experiment. Similar results were obtained from three independent experiments, using pmirGLO-16 as an internal control. NC, Negative control. (**D**,**E**) HEK-293T cells were transfected with miR-H2-3p mimics or NC at indicated final concentrations. After 24 h of transfection, total RNA was extracted and miR-H2-3p expression was determined by qRT-PCR (**D**) or RT-PCR (**E**). The values were normalized to U6 snoRNA and standardized to 1 in the NC groups. Data are the means ± SD (*n* = 3) from one representative experiment. Similar results were obtained from three independent experiments, using hsa-miR-16 (miR-16) as an internal control. (**F**–**H**) HEK-293T cells were transfected with miR-H2-3p mimics or NC at indicated final concentrations. After 24 h of transfection, total RNA was extracted and DDX41 mRNA expression was determined by qRT-PCR (**F**) or DDX41 protein by immunoblotting (**G**), and the relative density of DDX41 protein was determined using ImageJ software (**H**). The values were normalized to GAPDH and standardized to 100% in the NC groups (**F**), using GAPDH as a loading control (G and H). Data are the means ± SD (*n* = 3) from one representative experiment. Similar results were obtained from three independent experiments. * *P* < 0.05; ** *P* < 0.01; *** *P* < 0.001.

**Figure 3 viruses-11-00756-f003:**
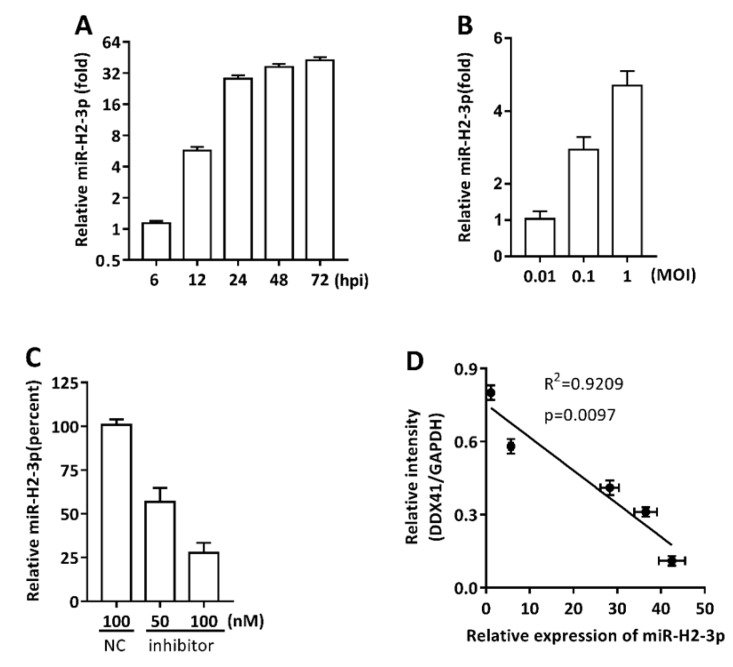
miR-H2-3p contributed to the downregulation of DDX41 during HSV-1 infection in THP-1 cells. (**A**–**C**) THP-1 cells were infected with HSV-1 at an MOI of 1 for indicated time point (**A**) or at an MOI of 0.01, 0.1, and 1 for 24 h (**B**), or transfected with miR-H2-3p inhibitor or negative control inhibitor at the indicated final concentration, and then infected with HSV-1 at an MOI of 1 for 24 h (**C**). The cells were harvested, and total RNA was extracted for determining the miR-H2-3p expression using qRT-PCR. The values were normalized to U6 sno RNA and standardized to 1 in 6 hpi (**A**) or an MOI of 0.01 (**B**) or NC treatment (**C**). (**D**) Analysis of correlation between miR-H2-3p expression (**A**) and DDX41 expression (**E**) during HSV-1 infection with an MOI of 1. R, Reliability; p, Pearson correlation coefficient. (**E** and **F**) THP-1 cells were infected with HSV-1 at an MOI of 1 or mock (Mock) for indicated time periods. The level of DDX41 protein was determined by immunoblotting (**E**) and the relative density of DDX41 protein using ImageJ software (**F**). The values were normalized to GAPDH, using GAPDH as a loading control. (**G**–**J**) THP-1 cells were transfected with miR-H2-3p mimics or negative control mimics at the final concentration of 100 nM or miR-H2-3p inhibitor or negative control inhibitor at the final concentration of 100 nM, and then infected with HSV-1 at an MOI of 1 for 24 h. The level of DDX41 protein was determined by immunoblotting (**G**,**I**) and relative density of DDX41 protein using ImageJ software (**H**,**J**). The values were normalized to GAPDH, using GAPDH as a loading control. (**K**) Summary for panel of F, H, and J. Data are the means ± SD (*n* = 3) from one representative experiment. Similar results were obtained from three independent experiments. * *P* < 0.05; ** *P* < 0.01.

**Figure 4 viruses-11-00756-f004:**
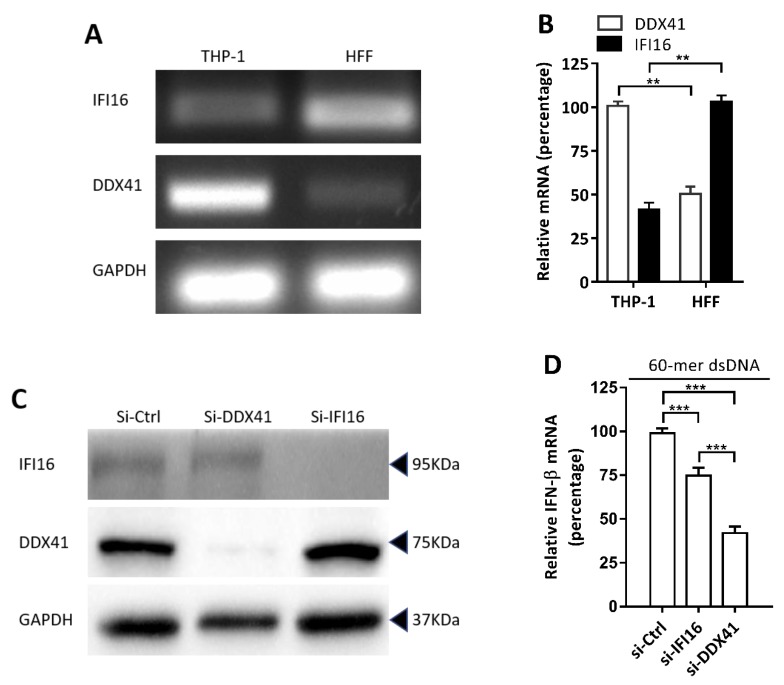
DDX41 was a crucial DNA sensor in THP-1 cells. (**A** and **B**) Total RNA was extracted from THP-1 or HFF cells and used for determining the expression of IFI16 and DDX41 by RT-PCR (**A**) and qRT-PCR (**B**), respectively. GAPDH was used as an internal control. (**C**) THP-1 cells were transfected with siRNA (final concentration of 50 nM) for the knockdown of DDX41 and IFI16 or nonspecific control (si-Ctrl), as indicated. After 24 h of transfection, the effects of knockdown of DDX41 and IFI16 were evaluated by immunoblotting, using GAPDH as a loading control. (**D**) THP-1 cells were treated with nonspecific control siRNA (si-Ctrl) or siRNA against IFI16 or DDX41, as indicated. After 24 h of treatment, THP-1 cells were transfected with HSV-1 60-mer dsDNA (final concentration of 1 µg/mL) for 8 h, and then the IFN-β mRNA level was determined by qRT-PCR. The values were normalized to GAPDH and standardized to 100% in the control groups, as indicated. Data are the means ± SD (*n* = 3) from one representative experiment. Similar results were obtained from three independent experiments. ** *P* < 0.01; *** *P* < 0.001.

**Figure 5 viruses-11-00756-f005:**
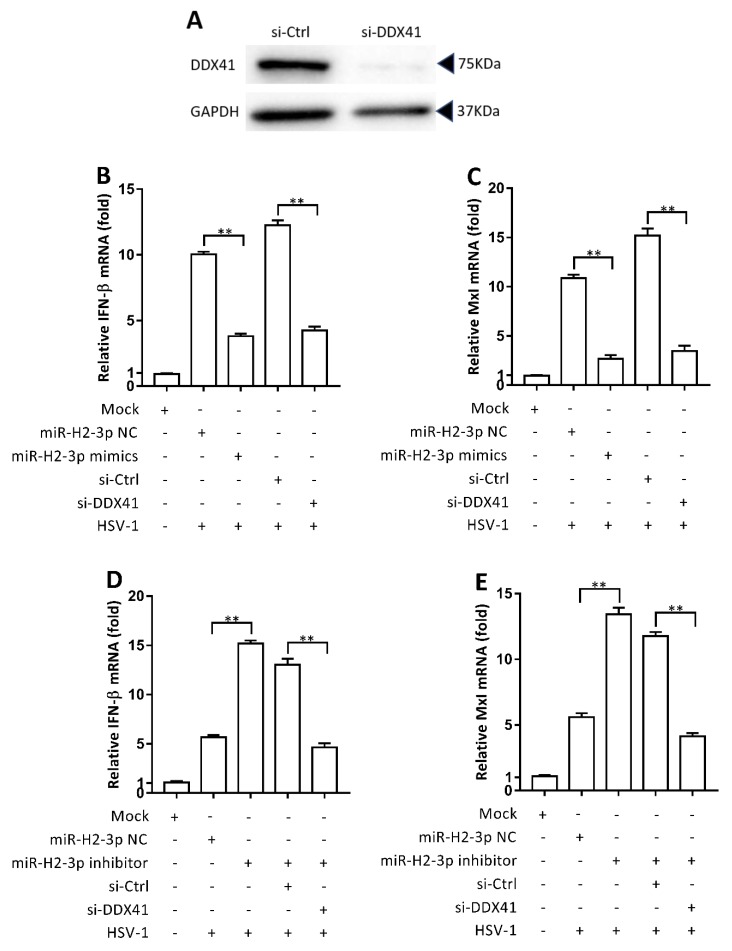
miR-H2-3p downregulated HSV-1-triggered IFN-β and ISG expression by targeting DDX41 directly. (**A**) THP-1 cells were transfected with siRNA (final concentration of 50 nM) for the knockdown of DDX41 or nonspecific control (si-Ctrl), as indicated. After 24 h of transfection, the effect of knockdown of DDX41 was evaluated by immunoblotting, using GAPDH as a loading control. (**B**,**C**) THP-1 cells were transfected with miR-H2-3p mimics or negative control (NC) (final concentration of 100 nM), siRNA against DDX41, or control (Ctrl) (final concentration of 50 nM), as indicated. After 24 h of transfection, THP-1 cells were infected with HSV-1 at an MOI of 1 or mock (Mock) for 24 h. The IFN-β (**B**) and MxI (**C**) expression levels were determined by qRT-PCR. The values were normalized to GAPDH. (**D**,**E**) THP-1 cells were transfected with miR-H2-3p inhibitor or miR-H2-3p NC (final concentration of 100 nM) and siRNA against DDX41 or control (Ctrl) (final concentration of 50 nM), as indicated. After 24 h of transfection, THP-1 cells were infected with HSV-1 at an MOI of 1 or mock (Mock) for 24 h. The IFN-β (**D**) and MxI (**E**) expression levels were determined by qRT-PCR, and the values were normalized to GAPDH. All data are the means ± SD (*n* = 3) from one representative experiment. Similar results were obtained from three independent experiments. ** *P* < 0.01.

**Figure 6 viruses-11-00756-f006:**
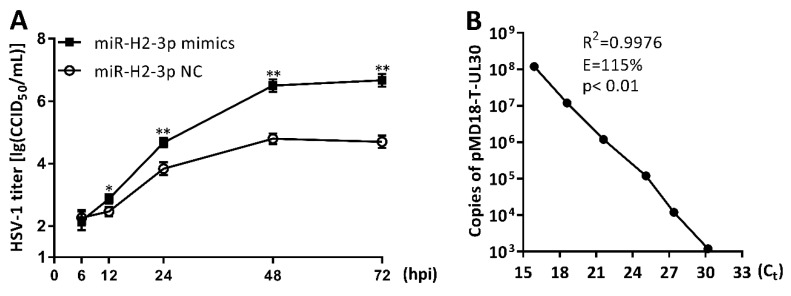
miR-H2-3p promoted HSV-1 replication and gene expression in THP-1 cells. (**A**) THP-1 cells were transfected with miR-H2-3p mimics or negative control (NC) (final concentration of 100 nM) for 24 h, and then infected with HSV-1 at an MOI of 1 for 6, 12, 24, 48, and 72 h, as indicated. The viral titers in the culture supernatant were determined by the CCID_50_ method. (**B**) Constructed pMD18-T-UL30 plasmids were subjected to 10 times series dilution and used as a template to amplify the viral *UL30* gene fragment by qPCR. Linear regression analysis with plasmid copy numbers and cycle threshold (C_t_) was performed. R, Reliability; E, amplification efficiency; p, Pearson correlation coefficient. (**C**) THP-1 cells were treated as in (**A**), and the viral genome copy numbers in THP-1 cells were determined by qPCR to amplify the viral *UL30* gene fragment. (**D**) THP-1 cells were co-transfected with miR-H2-3p mimics and negative control (NC) at various final concentrations for 24 h, as indicated, and then infected with HSV-1 at an MOI of 1 for 24 h. THP-1 cells were harvested, and total RNA was extracted. The expression of TK, gC, and ICP0 was determined by qRT-PCR, and the values were normalized to GAPDH. Data are the means ± SD (*n* = 3) from one representative experiment. Similar results were obtained from three independent experiments. * *P* < 0.05; ** *P* < 0.01; and *** *P* < 0.001.

**Figure 7 viruses-11-00756-f007:**
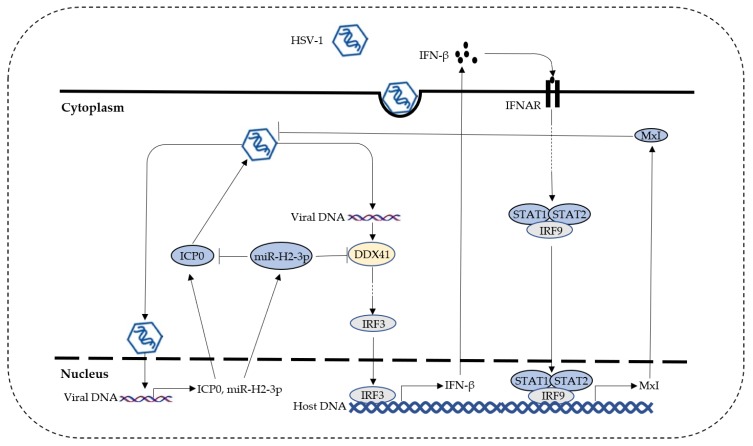
Schematic regulatory relationship involving HSV-1, ICP0, miR-H2-3p, DDX41, and MxI. HSV-1-infected host cells produce viral protein ICP0 and miR-H2-3p. ICP0 can promote viral replication in the early stage, while miR-H2-3p can repress the expression of ICP0 and DDX41. Meanwhile, viral DNA released into the cytoplasm is recognized by DDX41, triggering the production of IFN-β. IFN-β-induced antiviral effector protein MxI inhibits HSV-1 replication. The miR-H2-3p antagonizes DDX41-mediated antiviral innate immunity to escape elimination.

**Table 1 viruses-11-00756-t001:** Primers used for plasmid construction.

Gene	Primers (5′→3′)
*DDX41*	Forward: GAGCTCGCCGACAGTCTTCCCTTCTC
	Reverse: TCTAGATGGGCTAGAGGTTTGGGCTT
*IFI16*	Forward: GAGCTCAATCTGGATGTCATTGACGA
	Reverse: TCTAGAATGTTTATTTATTTTTTCAA
*DAI*	Forward: GAGCTCGCCGACAGTCTTCCCTTCTC
	Reverse: TCTAGAATGGTTGGTTTATTTTTTAT
*cGAS*	Forward: GAGCTCGATTGTATTTTTAGAAAGAT
	Reverse: TCTAGAGTGAGCCACAGCGTCT
*STING*	Forward: GAGCTCGACCCAGGGTCACCAGGCCA
	Reverse: TCTAGACCGTTGAGAAAGGGGTGGAG
*TRIF*	Forward: GAGCTCCCGCGTGTCCTTGCCTGACC
	Reverse: TCTAGAGCTGCGGAAGCATTCAATAA
*TBK1*	Forward: GAGCTCTTTCTAATAGAAGTTTAAGA
	Reverse: TCTAGATTCCATGAAATAGTAAGCAG
*TLR9*	Forward: GAGCTCCCGTGAGCCGGAATCCTGCA
	Reverse: TCTAGATGCTCTGTGTCAGGTGTGGG
*IRF*3	Forward: GAGCTCGCCCTCGCTCCTCATGGTGT
	Reverse: TCTAGATTTGATCATAGCAGGAACCA
*IRF7*	Forward: GAGCTCAACCCAGTCTAATGAGAACT
	Reverse: TCTAGATGTTCTGGAGTTCTTTTTAT
*IRF9*	Forward: GAGCTCAGCCTGGGGGACCCATCTTC
	Reverse: TCTAGAAAGTGCTTGGCTTTAGAGTT
*UL30*	Forward: CATCACCGACCCGGAGAGGGAC
	Reverse: GGGCCAGGCGCTTGTTGGTGTA
hsa-miR-16	Sense: CGCCAATATTTACGTGCTGCTA
	Antisense: CTAGTAGCAGCACGTAAATATTGGCGAGCT

**Table 2 viruses-11-00756-t002:** The siRNA sequences.

Gene	Sequences (5′→3′)
*DDX41*	Sense: GAUGGCUAAGGGCAUUACGTT
	Antisense: CGUAAUGCCCUUAGCCAUCTT
*IFI16*	Sense: TAGCGTTTCTGGAGATTACAATT
	Antisense: UUGUAAUCUCCAGAAACGCUATT
Scramble Control	Sense: CUAAUCAGGCGAUGAAUGUTT
	Antisense: ACAUUCAUCGCCUGAUUAGTT

**Table 3 viruses-11-00756-t003:** The miRNA mimics and inhibitor sequences.

Gene	Sequences (5′→3′)
miR-H2-3p mimics	Sense: CCUGAGCCAGGGACGAGUGCGACU
	Antisense: UCGCACUCGUCCCUGGCUCAGGUU
miR-H2-3p mimics NC	Sense: UUCUCCGAACGUGUCACGUTT
	Antisense: ACGUGACACGUUCGGAGAATT
miR-H2-3p inhibitors	AGUCGCACUCGUCCCUGGCUCAGG
miR-H2-3p inhibitors NC	CAGUACUUUUGUGUAGUACAA

**Table 4 viruses-11-00756-t004:** The RT-PCR primers.

Gene	Primers (5′→3′)
*DDX41*	Forward: GACACTGGTGTTCACGTTGC
	Reverse: CGAGGGGCAGATGATGAGTC
*IFI16*	Forward: AGAAACAATGACCCCAAGAGC
	Reverse: CTTGGTGAAGAAACTGCTGGAT
*IFN-β*	Forward: CCAACAAGTGTCTCCTCCAA
	Reverse: ATAGTCTCATTCCAGCCAGT
*MxI*	Forward: CTGAAGGAGCGGAGCGACAC
	Reverse: TTAACCGGGGAACTGGGCAA
*GAPDH*	Forward: CATGTTCGTCATGGGTGTGAACCA
	Reverse: AGTGATGGCATGGACTGTGGTCAT
*TK*	Forward: GATGGGGAAAACCACCACCA
	Reverse: TGTTCGCGATTGTCTCGGAA
*gC*	Forward: CCAACCTAACGGACCCCCAC
	Reverse: GCGTCACCTCGCCGATAATC
*ICP0*	Forward: ACACGGCAGCAGTAACACCA
	Reverse: TCCGCTTCAACAACCCCAAC
*miR-H2-3p*	RT: GTCGTATCCAGTGCAGGGTCCGAGGTATTCGCACTGGATACGACAGTCGCA
	Forward: CCCTGAGCCAGGGACGAG
	Reverse: CCAGTGCAGGGTCCGAGGTA
*U6 snoRNA*	Forward: GCTCGCTTCGGCAGCACATA
	Reverse: TGGAACGCTTCACGAATTTG

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
