# Peer review of "Herpes Simplex Virus Type 1–Encoded miR-H2-3p Manipulates Cytosolic DNA–Stimulated Antiviral Innate Immune Response by Targeting DDX41"

_viruses, 2019, doi:10.3390/v11080756_

Round 1

Reviewer 1 Report

Qihan Li and colleagues report that HSV-1 expresses a miRNA (H2-3p) that targets DDX41, a cytosolic DNA sensor that functions as an innate response product during lytic viral infection. Transfection of miR-H2-3p reduces DDX41 mRNA and protein and IFN-B and MxI that are  also reduced. Since little is known about HSV miRNAs, this is an interesting paper and speaks to the many ways this virus can interfere with innate immunity. The impact on DDX41 by mirH2-3P appears to be virus strain and situationally dependent based on other studies and unfortunately this report does not resolve these inconsistencies. It is unclear how mirH2 impedes ICP0 activity that seems counterproductive to aiding virus replication. Nevertheless, the experiments are well controlled and provide a consistent message.

Comments.

Throughout  this paper, there is a very longwinded accounting of everything known      about viruses and miRNA. Besides being largely irrelevant to this report, it makes reading the paper a laborious process and obscures the important  points. Much of the introduction and discussion belong in a review. For example, the first two paragraphs of the Introduction, the discussion of ICP0 and the first 3 paragraphs of the Discussion should be removed. The organization of the Discussion is not quite logical and last two paragraphs need rewriting and blending. The authors need to stick to the point.

There are  minor grammatical mistakes throughout such as plurals, verb tenses etc.  and this can be corrected by an English-speaking  editor.

Figure 1 A and B are unnecessary and picking 24 h for analysis of miRNA activity is  not the best choice since early activity would be required to impact virus replication efficiency. At 24 hours, the virus has easily completed the replication cycle and viral gene expression is no longer regulated. A more  relevant experiment might be to look for the expression and effects of the  mir H2-3p in cultures of primary ganglion neurons in a model of viral      latency where this mir may play a role. While DDX41 is impacted in their luciferase assay, there is a lot of “nonspecific” affects on a variety of gene products and is not remarkably different than the reduction in DAI and TRIF for example. The location of the p-values seems to cover all the tested genes. In general, the relevant expression of HSV miRNAs is likely limited to latency particualry since they are encoded in the latency region of the genome and most relevant to the virus life cycle in the host.

The      experiments in Fig. 2 are of significant interest although lanes 3 and 4 of G seem to be of lower molecular size. What’s the explanation for this?

The authors argue that mirH2 plays a role in blocking an aspect of innate immunity but      there are no quantitative virus production assays to demonstrate this that depend on virus expression of mirH2. Transfections can be worthwhile, however, the level of expression is quite high. A better and more convincing tact would be to mutate the mir coding sequence in the virus and show that replication is impaired. How important is mirH2 to improving      virus replication and under what circumstances. My guess is that the HSV mirs play a role in either the establishment of latency or virus reactivation from latency and the latter appears relevant for mirH2-3P. Since Fig 6C shows a small but significant impact on viral DNA levels at late time points, the impact of mirH2 may be subtle and important under special circumstances.

An interesting outcome showed that IFNb/MxI is repressed by mirH2 overexpression. Why wasn’t this included in the scheme in Fig. 7. Could the authors discuss the possible mechanism for this repressive activity.

Reviewer 2 Report

Duan and colleagues demonstrated in this work that miRNA-H2-3p encoded by HSV1 can allow the virus escaping  the cell innate immune response by targeting DDX41 at the transcriptional level.

Demonstration of the link between miRNA expression, by transfection or infection, and inhibition of DDX41 protein expression is convincing. The role of DDX41 in sensing HSV1 infection is also well demonstrated. Finally, the authors show that miRNA-induced immunomodulation promotes replication and transcription of HSV1.

Some clarifications and improvements may still be made to the paper before publication:

-          In general, English must be reviewed and the paper carefully edited.

-          In the title of the first chapter of the results, it is not the activity of DDX41 that is modulated by HSV1 but its expression. Please correct. In Figure 1B, I would show the expression of miR at different times and different MOI rather than the CPE. In Figures 1C and 1D, it is not clear for me if only DDX41 expression is significantly reduced compared to the empty vector or if all the innate immune cells are also inhibited.

-          In Figure 4, why did the authors test the respective role of IFI16 and DDX41 in the sensing of viral infection after transfection of viral DNA and not in the context of the infection?

-          In Figure 6, what is the physiological relevance of adding miR-H2-3p mimics to viral replication? Can we reach the same concentrations of miR in the normal context of the infection? Why not have tested the impact of miR inhibitors on the replication of HSV1?

-          The discussion is difficult to read and poorly reflects the results of the study. It could be significantly improved. The link between the expression of miR and ICP0 lacks clarity. Overexpression of TK and GC in the presence of miR is not discussed. What are the physiopathological consequences of these different phenomena? Figure 7 shows new partners such as Trim21 and hsa-miR-138 that do not appear in the experiments and proposes an interaction scheme that is not fully demonstrated by the work done. Finally, it is important to put into perspective the inhibition of DDX41 in the context of infection. Why does not the virus escape the immune response? What are the consequences of overexpression of this miR on other proteins involved in the sensing of HSV infection (see comment about Figure 1) ?

Minor comments :

-          Virus production : why three-time freezing and thawing since HSV-1 is an envelopped virus only infectious if collected from the supernatant ?

-          Materials and Methods : please, put all the primers in a table.

-          Please, define miR-H2-3p mimics. Are they synthetic miR ?

-          The numbering in Figure 2 does not correspond to the legend and the text.

-          Define U6 sno RNA and write it in the same way in both page 6 lines 261-262 and page 14 line 427.

-          The sentences 517 to 521 page 17 are not clear. Please, rephrased.

-          Discussion, page 19: It is not clear whether hsa-miR comes from the host cell or the virus.

-          Define MICB line 570 page 19.

Author Response

Response to Reviewer 2 Comments
Comments.
Duan and colleagues demonstrated in this work that miRNA-H2-3p encoded by HSV1 can allow the virus escaping the cell innate immune response by targeting DDX41 at the transcriptional level.
Demonstration of the link between miRNA expression, by transfection or infection, and inhibition of DDX41 protein expression is convincing. The role of DDX41 in sensing HSV1 infection is also well demonstrated. Finally, the authors show that miRNA-induced immunomodulation promotes replication and transcription of HSV1.
Some clarifications and improvements may still be made to the paper before publication:
- In general, English must be reviewed and the paper carefully edited.
- In the title of the first chapter of the results, it is not the activity of DDX41 that is modulated by HSV1 but its expression. Please correct. In Figure 1B, I would show the expression of miR at different times and different MOI rather than the CPE. In Figures 1C and 1D, it is not clear for me if only DDX41 expression is significantly reduced compared to the empty vector or if all the innate immune cells are also inhibited.
- In Figure 4, why did the authors test the respective role of IFI16 and DDX41 in the sensing of viral infection after transfection of viral DNA and not in the context of the infection?
- In Figure 6, what is the physiological relevance of adding miR-H2-3p mimics to viral replication? Can we reach the same concentrations of miR in the normal context of the infection? Why not have tested the impact of miR inhibitors on the replication of HSV1?
- The discussion is difficult to read and poorly reflects the results of the study. It could be significantly improved. The link between the expression of miR and ICP0 lacks clarity. Overexpression of TK and GC in the presence of miR is not discussed. What are the physiopathological consequences of these different phenomena? Figure 7 shows new partners such as Trim21 and hsa-miR-138 that do not appear in the experiments and proposes an interaction scheme that is not fully demonstrated by the work done. Finally, it is important to put into perspective the inhibition of DDX41 in the context of infection. Why does not the virus escape the immune response? What
are the consequences of overexpression of this miR on other proteins involved in the sensing of HSV infection (see comment about Figure 1)?
Minor comments:
- Virus production: why three-time freezing and thawing since HSV-1 is an envelopped virus only infectious if collected from the supernatant?
- Materials and Methods: please, put all the primers in a table.
- Please, define miR-H2-3p mimics. Are they synthetic miR?
- The numbering in Figure 2 does not correspond to the legend and the text.
- Define U6 sno RNA and write it in the same way in both page 6 lines 261-262 and page 14 line 427.
- The sentences 517 to 521 page 17 are not clear. Please, rephrased.
- Discussion, page 19: It is not clear whether hsa-miR comes from the host cell or the virus.
- Define MICB line 570 page 19.

Point 1: Duan and colleagues demonstrated in this work that miRNA-H2-3p encoded by HSV1 can allow the virus escaping the cell innate immune response by targeting DDX41 at the transcriptional level.
Demonstration of the link between miRNA expression, by transfection or infection, and inhibition of DDX41 protein expression is convincing. The role of DDX41 in sensing HSV1 infection is also well demonstrated. Finally, the authors show that miRNA-induced immunomodulation promotes replication and transcription of HSV1.
Some clarifications and improvements may still be made to the paper before publication:

Response 1: Than you, we have made adjustments to the logical relationship, the retouching of the language and the rewriting of the introduction and discussion.

Point 2: In general, English must be reviewed and the paper carefully edited.

Response 2: Thank you, we have corrected grammatical mistakes and polished by professional English-speaking team.

Point 3: In the title of the first chapter of the results, it is not the activity of DDX41 that is modulated by HSV1 but its expression. Please correct. In Figure 1B, I would show the expression of miR at different times and different MOI rather than the CPE. In Figures 1C and 1D, it is not clear for me if only DDX41 expression is significantly reduced compared to the empty vector or if all the innate immune cells are also inhibited.

Response 3: Thank you, we have corrected. We have changed “activity of DDX41” into “expression of DDX41”. In Figure 1B, the purpose is to determine the infection status of the virus. Firstly, virally encoded miRNAs are abundantly expressed in the late stages of infection, so to play its biological function should be in the late stages, secondly, for subsequent experiments, the number of cells cannot be reduced too much due to the infection process of the virus, therefore, CPE observation is a better choice. We show the expression of miR-H2-3p in Figure 3A and 3B. In Figure 1C and 1D, this was caused by our carelessness, we have corrected it.

Point 4: In Figure 4, why did the authors test the respective role of IFI16 and DDX41 in the sensing of viral infection after transfection of viral DNA and not in the context of the infection?

Response 4: Thank you, viral infections will produce more complex causal relationship than viral DNA, so we chose a simple way to prove its importance.

Point 5: In Figure 6, what is the physiological relevance of adding miR-H2-3p mimics to viral replication? Can we reach the same concentrations of miR in the normal context of the infection? Why not have tested the impact of miR inhibitors on the replication of HSV1?

Response 5: Thank you, our aim is to prove that miR-H2-3p can promote viral replication. The concentration of transfection should be much higher than that of normal infection, but this is a common research method. The efficiency of miRNAs inhibitor is relatively low, if the concentration of inhibitors is increased, it will have a serious impact on cells growth as well as viral replication. In contrast, miRNAs mimics achieves overexpression at lower concentrations. Maybe using miRNAs sponge can achieve better inhibition.

Point 6: The discussion is difficult to read and poorly reflects the results of the study. It could be significantly improved. The link between the expression of miR and ICP0 lacks clarity. Overexpression of TK and GC in the presence of miR is not discussed. What are the physiopathological consequences of these different phenomena? Figure 7 shows new partners such as Trim21 and hsa-miR-138 that do not appear in the experiments and proposes an interaction scheme that is not fully demonstrated by the work done. Finally, it is important to put into perspective the inhibition of DDX41 in the context of infection. Why does not the virus escape the immune response? What are the consequences of overexpression of this miR on other proteins involved in the sensing of HSV infection (see comment about Figure 1)?

Response 6: Thank you, we have rewritten the discussion, especially added the discussion about TK and gC; miR-H2-3p, ICP0 and HSV-1 replication. We have changed the schematic regulatory relationship involving DDX41, IFN-βand MxI in Figure 7 and added relative discussion.

Point 7: Virus production: why three-time freezing and thawing since HSV-1 is an envelopped virus only infectious if collected from the supernatant?

Response 7: Thank you, this is a method of harvesting viruses commonly used in our laboratory, the purpose is to release virions from cells to increase the titer of the virus. Perhaps this treatment has its shortcomings, the obvious one is that repeated freeze-thaw cycles may make the virus lose its ability to infect.

Point 8: Materials and Methods: please, put all the primers in a table.

Response 8: Thank you, we have put all the primers and oligonucleotides in table.

Point 9: Please, define miR-H2-3p mimics. Are they synthetic miR?

Response 9: Thank you, miR-H2-3p is a synthetic double-stranded miRNA analog, overexpression of specific miRNA after transfection of cells.

Point 10: The numbering in Figure 2 does not correspond to the legend and the text.
Response 10: Thank you, this was caused by our carelessness, we have corrected it.

Point 11: Define U6 snoRNA and write it in the same way in both page 6 lines 261-262 and page 14 line 427.
Response 11: Thank you, U6 snoRNA is a small nucleolar RNA (snoRNA) in the nucleus, used as an internal reference for quantifying non-coding RNA. We have rewritten it in the same way in manuscript.

Point 12: The sentences 517 to 521 page 17 are not clear. Please, rephrased.
Response 12: Thank you, we have deleted these sentences and added relative part in discussion.

Point 13: Discussion, page 19: It is not clear whether hsa-miR comes from the host cell or the virus.

Response 13: Thank you, hsa- is comes from the host cell, especially from (Homo sapiens) human.

Point 14: Define MICB line 570 page 19.

Response 14: Thank you, in our revision, we have deleted relevant content. MICB gene encodes MHC class I polypeptide-related sequence B protein.

Round 2

Reviewer 1 Report

nice rewrite--greatly improved readability

Reviewer 2 Report

The authors responded satisfactorily to the remarks made during the first review. I have no other comments.